# Genome-wide association studies reveal novel loci associated with pyrethroid and organophosphate resistance in *Anopheles gambiae* and *Anopheles coluzzii*

Eric R. Lucas [1] ✉, Sanjay C. Nagi[1], Alexander Egyir-Yawson[2], John Essandoh [2], Samuel Dadzie [3], Joseph Chabi [3], Luc S. Djogbénou[4], Adandé A. Medjigbodo[4], Constant V. Edi[5], Guillaume K. Kétoh[6], Benjamin G. Koudou[5], Arjen E. Van't Hof [1,7], Emily J. Rippon[1], Dimitra Pipini[1], Nicholas J. Harding[8], Naomi A. Dyer [1], Louise T. Cerdeira[1], Chris S. Clarkson[9], Dominic P. Kwiatkowski [9,10], Alistair Miles[9], Martin J. Donnelly [1,9] ✉ & David Weetman[1] ✉

Resistance to insecticides in *Anopheles* mosquitoes threatens the effectiveness of malaria control, but the genetics of resistance are only partially understood. We performed a large scale multi-country genome-wide association study of resistance to two widely used insecticides: deltamethrin and pirimiphos-methyl, using sequencing data from *An. gambiae* and *An. coluzzii* from ten locations in West Africa. Resistance was highly multi-genic, multi-allelic and variable between populations. While the strongest and most consistent association with deltamethrin resistance came from *Cyp6aa1*, this was based on several independent copy number variants (CNVs) in *An. coluzzii*, and on a non-CNV haplotype in *An. gambiae*. For pirimiphos-methyl, signals included *Ace1*, cytochrome P450s, glutathione S-transferases and the *nAChR* target site of neonicotinoid insecticides. The regions around *Cyp9k1* and the *Tep* family of immune genes showed evidence of cross-resistance to both insecticides. These locally-varying, multi-allelic patterns highlight the challenges involved in genomic monitoring of resistance, and may form the basis for improved surveillance methods.

Vector-borne diseases such as malaria kill an estimated 700,000 people every year[1] but are vulnerable to methods that target the vectors that are essential for transmission. In malaria, the primary method of disease control remains the use of insecticides to kill the mosquitoes that transmit the disease[2]. Mirroring the evolution of drug resistance in pathogens, there has been widespread evolution of insecticide resistance in malaria mosquitoes[3], and understanding the genetic basis of this resistance is crucial for managing and informing malaria control

interventions. This applies both to current widely-used insecticides and new compounds coming to market, where the opportunity exists to understand the basis of resistance at the earliest stages of deployment.

The primary malaria vectors in Sub-Saharan Africa are *Anopheles gambiae* and *An. coluzzii*, two sister species of mosquito that have largely similar genomes and well-documented capacity for hybridisation and introgression[4–7]. Major mutations with large effects on

resistance, often at the insecticide's target site of action, have been discovered in these species, yet a large portion of the phenotypic variance in resistance remains unexplained[8]. This "residual" resistance involves a variety of detoxification mechanisms, such as increased expression or efficacy of genes that bind, metabolise or transport the insecticide. Identifying these mechanisms is challenging because the pool of genes potentially involved is much larger than that of the target site genes, and because modification of gene expression can occur in many ways.

Studies of residual resistance thus require large scale data, both in terms of genomic coverage and sample size. To this end, we set up the Genomics for African *Anopheles* Resistance Diagnostics (GAARD, https://www.anophelesgenomics.org) project, a collaboration to investigate the genomics of insecticide resistance through large scale whole genome sequencing. Here we investigate the genomic basis of resistance in West Africa to two insecticides widely used in malaria control. The first, deltamethrin, is a pyrethroid commonly used in insecticide-treated bednets (ITNs), which are the cornerstone of vector control[9]. Since ITNs have now been in circulation for many years, the mosquito populations in our study have a relatively long history of exposure. The second insecticide is pirimiphos-methyl (PM), an organophosphate that is deployed in the form of indoor residual spraying (IRS), where interior walls of buildings are coated to kill mosquitoes when they rest. Use of PM is more recent and more sporadic, as IRS is less widely implemented[10], making PM resistance less widespread than deltamethrin resistance (Fig. S1).

In this work, we conduct a large-scale, multi-country genome-wide association study (GWAS) of insecticide resistance in *An. gambiae* and *An. coluzzii*, testing mosquitoes from six different populations from four different West African countries for resistance against deltamethrin and PM. We find that resistance is highly multi-allelic, with different study populations showing different markers of resistance, even when the locus itself is the same, posing difficulties for genotypic monitoring programmes. We also make recommendations for sample size of future GWAS studies of insecticide resistance.

## Results
### Overview of data
We obtained sequencing data from 969 individual female mosquitoes across 10 sample sets (defined as samples from a single location of a given species phenotyped against one insecticides, Fig. 1 and Table 1). The phenotype of each individual was defined by whether they were alive (resistant) or dead (susceptible) after exposure to a given dose of insecticide.

Given that mosquitoes were derived from larval collections, we investigated whether our samples included close kin pairs, which could potentially introduce population stratification. Pairwise

**Table 1 | Number of samples sequenced in each of the sample sets**

| Location (country) | Species | Insecticide | N dead/alive[a] | Final N dead/alive[b] |
|---|---|---|---|---|
| Aboisso (Côte d'Ivoire) | *An. gambiae* | PM | 5/33 | 5/32 |
| Avrankou (Benin) | *An. coluzzii* | Delta | 40/48 | 34/45 |
| Baguida (Togo) | *An. gambiae* | PM | 33/42 | 30/35 |
| Baguida (Togo) | *An. gambiae* | Delta | 43/61 | 34/54 |
| Korle-Bu (Ghana) | *An. coluzzii* | PM | 55/62 | 48/57 |
| Korle-Bu (Ghana) | *An. coluzzii* | Delta | 88/61 | 83/59 |
| Madina (Ghana) | *An. gambiae* | PM | 33/40 | 27/38 |
| Madina (Ghana) | *An. gambiae* | Delta | 48/79 | 37/64 |
| Obuasi (Ghana) | *An. gambiae* | PM | 56/62 | 50/54 |
| Obuasi (Ghana) | *An. gambiae* | Delta | 51/29 | 51/27 |

GPS coordinates of collection sites are given in Supplementary Data 1.
*PM* pirimiphos-methyl, *Delta* deltamethrin.
[a]Sample size after sequencing QC.
[b]Sample size after removal of siblings.

calculations of kinship confirmed the presence of full siblings. We aggregated individuals into full sib groups by considering that two individuals that share a full sibling are also full siblings, resulting in the identification of 99 sib groups containing a total of 238 individuals (max sib group size: 9; 73% of sib groups were of size 2). Samples from different locations were never found in the same sib group. Depending on the analysis (see methods), we either discarded all-but-one randomly chosen individual per sib group per sample set (thus removing 105 samples) or performed permutations in which we varied which individuals were discarded in each sib group.

### Known resistance SNPs
In *An. coluzzii* from Avrankou, *Cyp4j5*–43F ($P = 0.007$) and *Vgsc*–1527T ($P = 0.02$) were both associated with increased resistance to deltamethrin. *Vgsc*-402L (generated by either of two nucleotide variants: 402L(C) and 402L(T)) and *Vgsc*–1527T are completely linked, and thus *Vgsc*-1527T here refers to the haplotype carrying both SNPs. Since *Vgsc*–995F and *Vgsc*–402L/1527T are mutually exclusive, and wild-types are absent in our dataset (Fig. 2), the significant association of *Vgsc*-402L/1527T suggests that this haplotype provides higher resistance to deltamethrin than does *Vgsc*–995F. None of the established resistance markers were associated with deltamethrin resistance in any other populations.

*Ace1*–280S was highly significantly associated with resistance to PM in all populations ($P < 0.001$ in all cases) except Baguida. In two populations, we also found PM-associations with other markers: in *An. gambiae* from Baguida, *Vgsc*–1570Y was positively associated with resistance ($P = 0.01$), which is a surprising finding for a pyrethroid target site mutation, whilst in *An. coluzzii* from Korle-Bu, *Vgsc*–995F ($P = 0.04$) and *Gste2*–114T ($P = 0.02$) were negatively associated with PM resistance.

### Copy number variants
Mutations that increase the number of genomic copies of a gene, known as copy number variants (CNVs), have been shown to be commonly associated with insecticide resistance in insects, including *Anopheles* mosquitoes[11]. We identified CNVs in genes associated with metabolic resistance, using both gene copy number (using sequencing coverage to estimate the number of copies of each gene) and screening for known *An. gambiae* CNV alleles (CNVs for which the precise start and end points have been previously identified[12, 13]). We found CNVs in *Cyp6aa1/2* and the *Gste* cluster at high frequency in *An. coluzzii* populations, while CNVs in *Cyp9k1* were far more common in *An. gambiae* (Figs. 3 and S2).

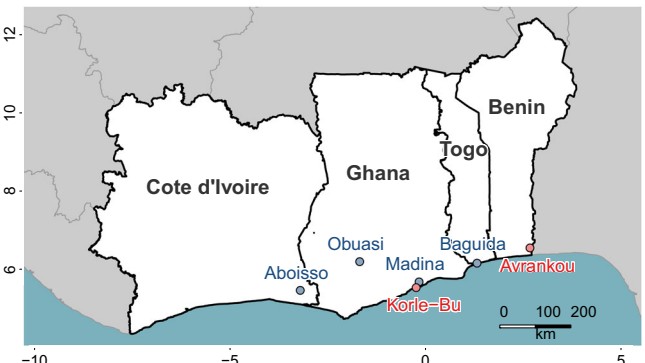

**Fig. 1 | Sampling locations for the study.** From each site, samples used for whole genome sequencing were *An. gambiae* (blue) or *An. coluzzii* (red). Axes show latitude and longitude.

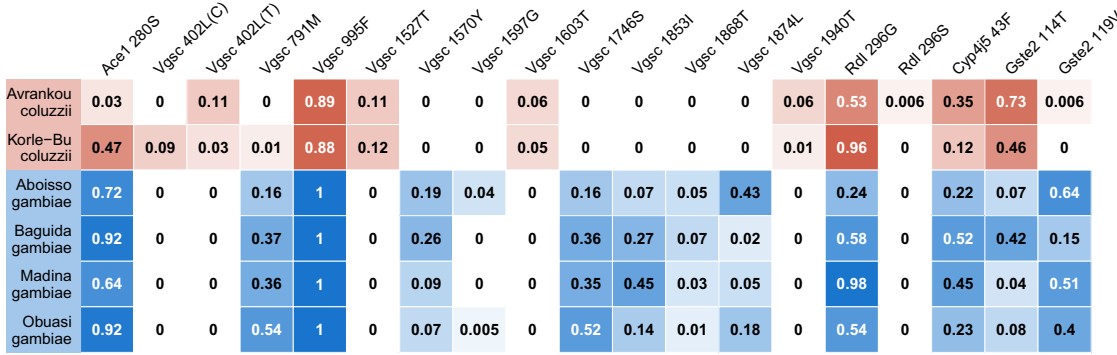

**Fig. 2 | Allele frequencies for SNPs in known resistance loci.** No wild-type hap-lotypes were found in *Vgsc*, with all samples carrying either the 995F mutation (100% of *An. gambiae* samples) or the 402L/1527T combination. Cells are colour-coded according to species (*An. gambiae* in blue, *An. coluzzii* in red) with darkness related to allele frequency. SNPs that were completely absent are not shown.

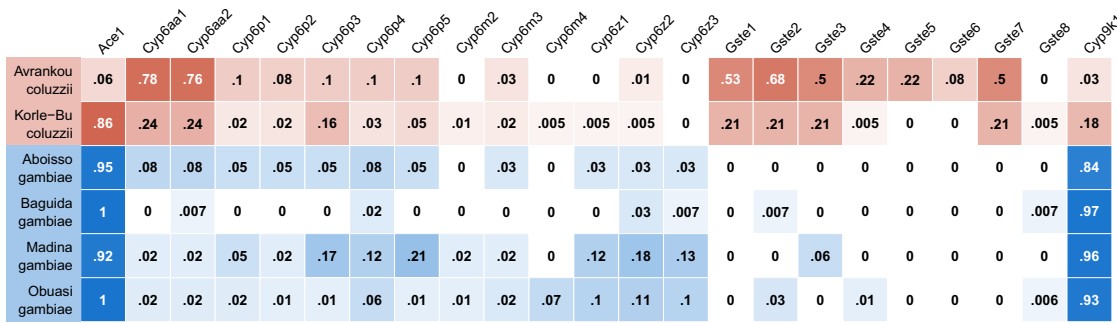

**Fig. 3 | Proportion of samples with increased copy number in genes from four major resistance loci** (*Ace1*, the *Cyp6aa/Cyp6p* cluster, the *Gste2* cluster and *Cyp9k1*). Cells are colour-coded according to species (*An. gambiae* in blue, *An. coluzzii* in red) with darkness related to proportion.

Copy number of *Cyp6aa1* was positively associated with delta-methrin resistance in *An. coluzzii*. This was significant in Korle-Bu ($P = 0.01$), with a trend in the same direction in Avrankou ($P = 0.07$). CNVs encompassing *Cyp6p3* and *Cyp6p5* exist at appreciable frequencies in *An. gambiae* from Madina (17% and 21% of samples showing increased copy number respectively) but with no significant association with resistance.

When breaking down the *Cyp9k1* CNVs into distinct alleles (Fig. S2), it is apparent that one allele (Cyp9k1_Dup10) is shared by both *An. gambiae* and *An. coluzzii*. Haplotype clustering analysis showed that the CNV in both species was present on the same genetic background (Fig. S3), which was nested within the *An. coluzzii* part of the haplotype tree, indicating that the mutation spread through introgression from *An. coluzzii* to *An. gambiae*. There was however no association between *Cyp9k1* copy number and resistance to either deltamethrin or PM.

The CNV in *Ace1*, known to be associated with PM resistance[6], was at high frequency (>85%) in all populations except *An. coluzzii* from Avrankou (Figs. 3 and S2), and *Ace1* copy number was strongly asso-ciated with resistance to PM in all populations ($P < 10^{-5}$ in all cases) except Baguida. No other genes showed a significant association between copy number and resistance.

### Windowed measures of differentiation/selection to identify genomic regions associated with resistance

We identified regions of the genome associated with phenotype within each sample set using three different metrics ($F_{ST}$, PBS and $\Delta H_{12}$) cal-culated in 1000 SNP windows (Figs. S4–S7). $F_{ST}$ is a measure of genetic differentiation between two groups of samples but does not indicate whether either group displays signals of selection, which is expected if the genetic difference is associated with resistance. Peaks of $F_{ST}$ were therefore investigated further to identify high-frequency haplotypes

significantly associated with resistance (Supplementary Data 2). PBS[14] is a measure of selection that is particularly effective at identifying recent selection from standing genetic variation. PBS identifies genomic regions showing greater evolutionary change in one group (here, the resistant samples) relative to a closely related group (susceptible sam-ples) and an outgroup. While originally designed to detect positive selection, it has also been used to detect phenotypic association[6]. $H_{12}$ is a measure used to detect genomic regions undergoing selective sweeps[15,16]. To identify regions in which swept haplotypes are more frequent in resistant compared to susceptible individuals, we calculated the difference in $H_{12}$ value between groups, which we refer to as $\Delta H_{12}$. Detailed breakdowns of all regions associated with resistance can be found in the Github repository https://github.com/vigg-lstm/GAARD_work/tree/main/supplementary/html_summaries/summary_files[17].

Based on these metrics, the cytochrome P450 *Cyp9k1* was fre-quently associated with resistance to both deltamethrin and PM, but curiously the association signal never localised to the gene itself (Figs. S4 and S8). We found signals on the telomeric side (Madina deltame-thrin, Madina PM, Obuasi PM) and the centromeric side (Avrankou deltamethrin, Obuasi deltamethrin, Baguida PM, Korle-Bu PM) or both (Korle-Bu deltamethrin), with distances from *Cyp9k1* ranging from 200 to 1200 Kbp. We also found recurrent, cross-insecticide signals in Thioester-containing protein (*Tep*) genes, with peaks in *Tep1* (AGAP010815) for Obuasi PM, and *Tep4* (AGAP010812) for Madina deltamethrin (Fig. S4).

For deltamethrin, the most consistent signal was at the *Cyp6aa/Cyp6p* gene cluster. At this locus, we found PBS and $F_{ST}$ peaks in *An. gambiae* from Obuasi and Madina, and a peak in $\Delta H_{12}$ in *An. coluzzii* from Korle-Bu. The strongest of these signals came from the Obuasi sample set, where the $F_{ST}$ peak spanned 17 windows spread over nearly 250 Kbp. The window covering the *Cyp6aa/Cyp6p* cluster

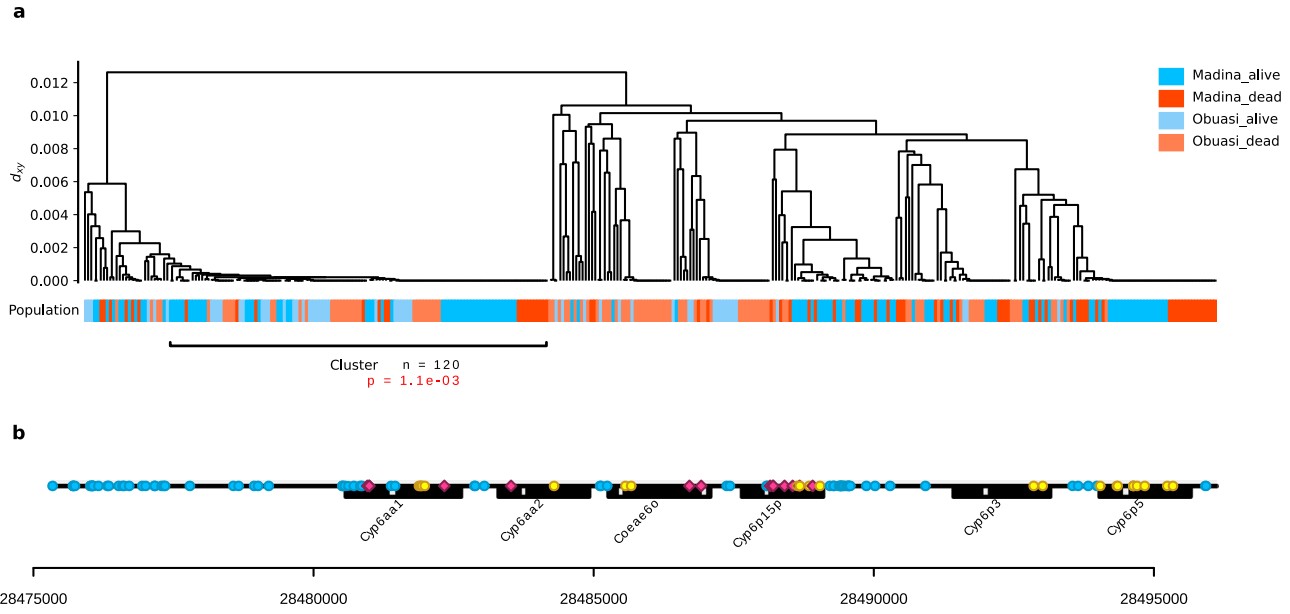

**Fig. 4 | A shared haplotype cluster around the *Cyp6aa/Cyp6p* cluster at high frequency in *An gambiae* from Madina and Obuasi is positively associated with resistance to deltamethrin. a** Hierarchical clustering dendrogram of haplotypes with leaves labelled by population and phenotype. The tree was cut at a height of $d_{xy} = 0.001$ to obtain haplotype clusters. **b** SNPs that were significantly more frequent on the haplotype than in the rest of the population (Fisher test, alpha = 0.001) are shown and labelled as non-synonymous (red), synonymous (yellow) and non-coding (blue, including UTRs).

contained a haplotype cluster that accounted for over a third of the sample set (56 out of 156 haplotypes) which was positively associated with resistance to deltamethrin ($P = 0.005$). In Madina, the peak consisted of two significant windows, the nearest one being around 6 Kbp away from the cluster. This window contained a haplotype group accounting for over half (109 out of 202) of the haplotypes in the sample set and was positively associated with resistance to deltamethrin ($P = 0.002$).

To determine whether the haplotypes driving the $F_{ST}$ signal in Obuasi and Madina were the same, we combined the deltamethrin-phenotyped samples from these two locations and re-analysed the significant window covering *Cyp6aa1*. The main haplotype cluster was indeed the same in Obuasi and Madina (Fig. 4) and the significance of association for the combined analysis was increased ($P = 0.001$). The haplotype cluster contained many SNPs (Fig. 4), of which four were non-synonymous in *Cyp6aa1* (2R positions 28480960:N501I, 28480993 + 28480994 (combine to make S490I) and 28482335:Y77F) and one in *Cyp6aa2* (28483525: M428V), but none of these SNPs were completely absent from the non-cluster haplotypes, suggesting that they may not be driving the sweep.

Several peaks were associated with genes from the superfamilies of metabolic genes typically associated with resistance (P450s, GSTs and carboxylesterases). We found PBS signals at *Cyp6m2* (Obuasi), *Cyp6ag1/2* (Avrankou) and *Cyp6aj1* (Madina) and an $F_{ST}$ signal near the carboxylesterase *Coe22933* in *An. coluzzii* from Korle-Bu (Fig. S4). In *An. coluzzii* from Avrankou, one $F_{ST}$ peak was adjacent to the P450 redox partner gene NADPH cytochrome P450 reductase (*Cpr*, AGAP000500) and contained a haplotype group positively associated with resistance ($P = 0.048$). There were also several independent signals in NADH dehydrogenase (ubiquinone) 1, with significant PBS peaks near beta subunit 2 (AGAP002630) in both *An. gambiae* from Madina and *An. coluzzii* from Korle-Bu. Furthermore, the PBS peak near *Cyp9k1* in Korle-Bu covers NADH dehydrogenase beta subunit 1 (AGAP000849), but the proximity to *Cyp9k1* makes it hard to determine whether this is coincidental.

For PM, we found strong peaks in the *Ace1* region in all populations except *An. gambiae* from Baguida (Fig. S4). Additionally, in *An. coluzzii* from Korle-Bu, a PBS and $F_{ST}$ peak that included the *Gste* gene family contained a haplotype cluster negatively associated with PM resistance ($P = 0.003$) and a small cluster ($n = 30$ haplotypes out of 210) positively associated with resistance ($P = 0.03$). There was also a significant peak in $\Delta H_{12}$ in *An. gambiae* from Madina at the *Gste* locus (Figs. S4 and S6).

In *An. gambiae* from Obuasi, the broadest peak consisted of 19 windows on chromosome 3L ranging from 30477135 to 30697180, which includes methuselah-like protein 7 (*Gprmthl7*, AGAP011643) and is around 23 kbp away from an odorant receptor *Or11* (AGAP011631). However, the haplotype cluster in these windows is negatively associated with PM resistance ($P < 0.0006$). Interestingly, there was also a $F_{ST}$/PBS peak of a single window on chromosome 3R, which contained the gene coding for the beta-2 subunit of nicotinic acetylcholine receptor (AGAP010057), the target site of neonicotinoid insecticides. The main haplotype in this region was positively associated with PM resistance ($P = 0.0008$) and contained a non-synonymous SNP (N418Y).

Some sample sets (Avrankou deltamethrin, Korle-Bu PM, Madina PM) also showed phenotypic associations for many SNPs within the 2La chromosomal inversion. Strong linkage disequilibrium and the large size of this inversion makes identification of the region responsible for the association unfeasible.

## GWAS analysis

As well as the windowed approach above, we implemented a SNP-wise association analysis across the genome in each sample set. Preliminary analysis identified significant associations that were later revealed to be an artefact of bacterial sequence data; we therefore repeated the analysis after removing SNPs associated with this contamination.

After false-discovery rate (FDR) control, no SNPs remained significant for deltamethrin resistance in any of the populations, while the PM sample sets either contained no significant SNPs (Baguida and Obuasi) or predominantly SNPs in the *Ace1* region (Korle-Bu and Madina). This likely reflects a combination of the effect size of resistance mutations, the sample sizes in our study, and the stringent false

discovery rate control when such a large number of tests are conducted. We reasoned that if mutations in a given region or cluster of genes are associated with resistance, several mutations in the same region may all contribute separately to resistance, or may lead to association of nearby variants through selective sweeps. We would then expect to see groups of clustered variants all associated with resistance. We therefore adopted an alternative approach in which we took the 1000 most highly significant SNPs in each sample set and looked for regions of the genome in which these SNPs were concentrated (100,000 bp windows that contained at least 10 SNPs from among the top 1000). For convenience, we refer to these as candidate SNPs. A summary of candidate SNPs can be obtained from the GitHub repository (https://github.com/vigg-lstm/GAARD_work/tree/main/supplementary/html_summaries/summary_files/gwas_summary.html)[17].

For deltamethrin, in *An. coluzzii* from Avrankou, the majority of the candidate SNPs were found across the 2La chromosomal inversion, mirroring what was found in the $F_{ST}$/PBS analysis. Aside from this, there were no candidate SNPs within gene sequences. In *An. gambiae* from Baguida, candidate SNPs were found in several regions of the genome, including a cluster of *Tep* genes (*Tep2* (AGAP008366), *Tep14* (AGAP008368) and *Tep15* (AGAP008364)), with a non-synonymous mutation in *Tep14*. Interestingly, we also found a window of candidate SNPs in *An. coluzzii* from Korle-Bu that was around 50 kbp from a different cluster of *Tep* genes, including *Tep1* (AGAP010815). We also found candidate SNPs around the cytochrome P450 *Cyp306a1*, including one within a splice site, and around the carboxylesterase *Coe22933*. In *An. gambiae* from Obuasi, the main signal came from the region around *Cyp6aa1*, as was found with the $F_{st}$ analysis.

For PM, in *An. coluzzii* from Korle-Bu and *An. gambiae* from Madina, all clusters of candidate SNPs were found in the *Ace1* region. In other populations, candidate SNPs were found in and around several clusters of possible detoxification genes (the carboxylesterases *Coeae2g-6g* in *An. gambiae* from Obuasi, and the P450s *Cyp12f1-f4*, *Cyp4d15-17*, *Cyp4k2*, *Cyp4ar1*, *Cyp4h19* and *Cyp4h24* in *An. gambiae* from Baguida). In Baguida, there was also a cluster of candidate SNPs covering a region around 39.2 Mb on chromosome 2L, containing many cuticular proteins (RR1 family AGAP006838-AGAP006867). In *An. gambiae* from Obuasi, as well as a group in *Ace1*, candidate SNPs were found near gustatory receptors (*Gr26* (AGAP006717) and *Gr27* (AGAP006716)) and a cluster of *Tep* genes (including five non-synonymous mutations in *Tep1*), as well as around *Gprmthl7* (including three non-synonymous mutations), as found in the $F_{ST}$ analysis.

## GWAS sample sizes

Although we found significant associations of individual SNPs in our analysis of established resistance markers for deltamethrin resistance, our agnostic SNP-level GWAS returned no markers passing FDR control. Using our results, we explored what sample size would have been required in order for these significant established markers to have been detected in the agnostic genome-wide analysis. Over 500 simulations, we found that a sample size of 300 (150 of each phenotype) would only have detected 11 or 25% of associated SNPs, depending on whether we modelled the observed allele frequencies of *Vgsc*_1527T or *Cyp4j5*_43F, respectively, in Avrankou. However, with a sample size of 500, this rose to 54 and 67%, respectively.

## Discussion

Overall, our results show different genomic signals of resistance to deltamethrin and PM, but with clear points of overlap. The pyrethroid deltamethrin is a common constituent of ITNs and has been a mainstay of vector control since the end of the twentieth century[18]. The first genetic marker of resistance against pyrethroids in *An. gambiae*,

discovered almost 25 years ago, was the target site mutation *kdr*-1014F, now referred to as *Vgsc*−995F[19]. Since then, this SNP has spread and massively increased in frequency[20], with at least five independent origins of the mutation, leaving the wild-type allele completely absent in our current dataset (collected in 2017). A recent decline in the frequency of *Vgsc*−995F in *An. coluzzii* is associated with the rise of an alternative resistance haplotype in the *Vgsc* gene, carrying the *Vgsc*−402L and *Vgsc*−1527T mutations instead[21]. While there is debate as to the relative benefits of *Vgsc*−995F vs *Vgsc*-402L/1527T, both provide target site resistance[22]. It is in this context of ubiquitous target site resistance to deltamethrin that our study was conducted, making it poised to address three questions: Is there a difference in the level of resistance conferred by *Vgsc*−995F and the *Vgsc*−402L/1527T haplotype? Do other SNPs around *Vgsc*−995F provide additional resistance to deltamethrin? And what other mutations in the genome help to explain the residual variation in resistance?

We found that *Vgsc*−402L/1527T was associated with significantly higher resistance to deltamethrin than *Vgsc*−995F. This is opposite to what has previously been found in laboratory colonies[22]. This difference may be explained by the effect of additional non-synonymous mutations on the *Vgsc*−995F haplotype background. The effect of most of these SNPs for resistance, likely to be through epistatic interaction with *Vgsc*−995F given their exclusive presence on this genetic background, have yet to be established. Such mutations were not exhaustively investigated in the laboratory colonies, and the relatively increased resistance of the *Vgsc*−995F haplotype in the colonies could thus be explained if this mutation was backed up by supporting SNPs elsewhere, or by differences in advantage between field and laboratory conditions.

Outside of the *Vgsc*, the main deltamethrin resistance-associated signals were in and around the *Cypaa*/*Cyp6p* cluster of cytochrome P450 genes. The importance of *Cyp6aa1* for deltamethrin resistance has been previously demonstrated in *An. gambiae* from East and Central Africa, where the rapidly spreading CNV Cyp6aap_Dup1 was found to be associated with resistance[23]. Similarly in our data, we found that copy number of *Cyp6aa1* was associated with resistance in at least one population of *An. coluzzii*. Unlike East Africa, where a single CNV allele dominates, we found at least 6 CNV alleles in our sample sets of *An. coluzzii*, and more are known from other data[12,13].

We found further signals of association between *Cyp6aa1* and deltamethrin resistance in two populations of *An. gambiae* (Madina and Obuasi). Interestingly, this signal was not associated with any CNV, suggesting an alternative mechanism of resistance. This non-CNV resistance haplotype, along with the large number of CNVs in the region, shows that metabolic resistance associated with *Cyp6aa1* is highly multi-allelic.

Other signals of association with deltamethrin resistance were found around the carboxylesterase *Coe22933* and the cytochrome P450s *Cyp9k1*, *Cyp6m2*, *Cyp6ag1/2*, *Cyp6aj1*, *Cyp306a1*, as well as *Cpr* (the obligatory cytochrome P450 electron donor, whose knockdown increases susceptibility to permethrin[24]). There was also a signal of association of deltamethrin resistance with the beta subcomplex of NADH dehydrogenase (ubiquinone), a large mitochondrial complex of the respiratory chain. We found such signals in populations from two species (*An. gambiae* from Madina and *An. coluzzii* from Korle-Bu), and possibly two regions of the genome (subunit 2 on chromosome 2R and subunit 1 on chromosome X), although the proximity of subunit 1 to *Cyp9k1* makes it hard to establish whether this is genuinely an independent signal of association. A previous study of deltamethrin resistance in a laboratory colony of *An. coluzzii* from Burkina Faso found that resistance was associated with elevated expression of genes with NADH dehydrogenase activity or involved in cellular respiration[25]. Our results suggest that this putative mechanism of resistance is present in other populations and species.

In contrast to deltamethrin, PM has been introduced more recently for mosquito control[10], although previous use of organophosphates as agricultural insecticides is likely to have led to prior exposure of at least some mosquito populations[26]. Furthermore, while *Vgsc* target site resistance to deltamethrin consists of presence/absence of a SNP, target site resistance to PM, through the *Ace1*−280S SNP, is moderated by the copy number of the CNV in *Ace1*[6]. This copy number is variable, and is likely to be highly mutable as recombination between CNV haplotypes leads to changes in copy number. For these reasons, variation in target site resistance persists, and our results show that this locus continues to dominate the association analyses in several populations.

Beyond *Ace1*, we found signals of association with PM resistance around cytochrome P450s (*Cyp4k1*, *Cyp12f1-4*, *Cyp4h19/24*, *Cyp9j5*) and carboxylesterases (*Coeae2g-6g*), as well as at the glutathione *S*-transferase epsilon (*Gste*) genes. Expression and sequence mutation in *Gste2* has been associated with resistance, primarily to DDT[27–29], but also to pyrethroids[28,30,31]. While direct evidence of an association with resistance to PM is lacking, GSTs are known to detoxify organophosphates and have frequently been associated with resistance to them in both *Anopheles* and *Aedes*[29,32,33].

Another intriguing signal was found at the beta-2 subunit of nicotinic acetylcholine receptor, where we found a haplotype with relatively strong association with PM resistance ($P = 0.0008$). This receptor is the target site for neonicotinoid insecticides[34] and is not known to be involved in resistance to PM, but its function is to respond to acetylcholine, the neurotransmitter that is broken down by *Ace1*. We can speculate that mutations in this receptor might in some way protect against the toxic accumulation of acetylcholine that occurs when *Ace1* is inhibited by PM.

Curiously, we also found marginally significant signals of association between PM resistance and SNPs in the *Vgsc* region (*Vgsc*−1570Y positively associated in Baguida and *Vgsc*−995F negatively associated in Korle-Bu). The negative association could be explained if the physiological cost of carrying the *Vgsc*−995F mutation compared to *Vgsc*-402L/1527 T[22,35] contributed to weakness in the face of stress. The effects of the *Vgsc*−1570Y mutation on fitness costs is unknown. If part of its effect is to reduce the fitness cost of *Vgsc*−995F, this may similarly explain its effect on PM resistance. These remain speculative arguments that require further investigation.

Given the importance of cross-resistance in the management of insecticide-based mosquito control strategies, regions found associated with resistance to both deltamethrin and PM warrant further discussion. The first was around *Cyp9k1*, a cytochrome P450 found in a region of strong selective sweep[16], that has been implicated in resistance to pyrethroids[36–38]. We found repeated signals of association of *Cyp9k1* with resistance to both deltamethrin and PM, although the signal was always several hundred, or even a thousand, Kbp away from *Cyp9k1* either side. This makes it difficult to confirm whether *Cyp9k1* is driving these signals. On the one hand, the known importance of *Cyp9k1* and the presence of significant windows on both sides, suggests that we may be seeing a broad signal of association that for some reason is weaker at the locus of the gene itself. Indeed, there are no known non-synonymous mutations associated with resistance in *Cyp9k1*, with the only evidence being haplotypic[36] and metabolic, and regulatory regions may be located far from the gene itself. On the other hand, genes such as NADH dehydrogenase and *Cyp4g17* were often close, or even within the significant windows. *Cyp4g17* has been found to be over-expressed in *An. coluzzii* from the Sahel resistant to deltamethrin[39], and the potential role of NADH dehydrogenase has been discussed above. The exact role that *Cyp9k1* and the genes around it play in resistance to both deltamethrin and PM remains poorly understood.

The other signals of association that were found for both insecticides were in *Tep* genes, key components of the arthropod innate immune system, with *Tep1* being known for its role in immunity against *Plasmodium*[40,41]. Similarly, a recent study of *An. coluzzii* in several regions of the Sahel found that *Tep1* was consistently over-expressed in mosquitoes resistant to deltamethrin[39]. It is intriguing to see *Tep1*, and other members of the *Tep* gene family, consistently associated with genomic signals of resistance to two insecticides, in both *An. gambiae* and *An. coluzzii*. While there is evidence that exposure to *Plasmodium* increases susceptibility to insecticides[42], inviting speculation that improving immunity could improve resistance, that study found that it is exposure to the parasite, rather than infection, that causes susceptibility.

When performing SNP-wise GWAS on the data, we found very few significant SNPs after false discovery rate control. In our targeted analyses of SNPs and CNVs in known resistance genes, the number of tests being conducted was considerably smaller and the range of *P* values that we found to be significant, with the exception of *Ace1*, would not come close to significance if FDR control was applied on a genomic scale such as in the GWAS. We conclude that the scale of the effect size for many mutations associated with resistance are such that larger sample sizes are required in order for their *P* values to stand out in a GWAS. Our simulations suggest that a sample size of 500, with a dataset of 7 million SNPS, of which 10 are associated with resistance, would detect 54% of SNPs that have the effect size that we observed for *Vgsc*_1527T in Avrankou, and 67% with the effect size observed for *Cyp4j5*_43F. This is an encouraging figure, and a reasonable one to achieve given current sequencing costs and throughput.

A striking result from our findings is that signals of non-target site resistance showed little consistency between our study sites. Apart from the *Cyp6aa1* haplotype associated with deltamethrin resistance in *An. gambiae* from Madina and Obuasi, few other signals were shared between populations, despite several repeated themes (*Cyp6aa1* resistance also present in *An. coluzzii*, *Tep* genes, P450s, carboxylesterases) and despite several documented instances of introgression between *An. gambiae* and *An. coluzzii* (*Vgsc*[4], *Ace1*[6], Rdl[5], *Cyp9k1* (this study)). These findings suggest that metabolic insecticide resistance in *An. gambiae* and *An. coluzzii* is highly multiallelic, with even nearby populations displaying different resistance mutations. This is in contrast to what has been observed with target site resistance, where various arrangements of a single SNP/CNV combination in *Ace1* dominate for PM resistance, while a few different mutations in *Vgsc* seem to be involved in deltamethrin resistance. It is also in contrast to *An. funestus*, where target site resistance has yet to be found in any populations, but where a few mutations associated with metabolic resistance have spread extensively in Africa[28,43–45]. It has been argued that resistance evolving naturally in the wild will tend to be monogenic, with a few genes of large effect being responsible for nearly all observed resistance, in contrast to resistance in artificially selected laboratory colonies[46]. This is because selective pressures imposed on lab colonies must necessarily be weak enough that a reasonable proportion of individuals survive to maintain the colony, allowing mutations with small effects to be selected. In contrast, insecticides deployed in the field, acting on a large population, can impose very high mortality such that only mutations at a very few loci of large effect on resistance are able to provide a selective advantage, although different mutations may well co-exist[46]. Our findings of many signals of association scattered over the genome seem to contradict this prediction. One possible explanation is that as initial mutations of large effect spread to fixation, as in the case of *Vgsc* and, in some populations, *Ace1*, so the mortality imposed by the insecticides decreases, leaving other mutations of small additive effect to spread on this background. The accuracy of the prediction may also depend on the method of insecticide application. Crops that are regularly sprayed with pesticides may maintain high levels of mortality, while bednets and sprayed walls are known to decline in potency over several years until net replacement or a new round of spraying[47–49]. Mosquitoes may

therefore be frequently exposed to lower doses of insecticides that select for many resistance loci of relatively small effect.

This multi-allelic nature of metabolic resistance presents a two-fold challenge for genetic monitoring programmes. First, if the markers of resistance are different in every population, SNP-focused marker panels will be less generalisable. This challenge can be overcome by turning to more general methods that target broader loci, such as amplicon sequencing, that remain agnostic as to the specific change that could occur within each broad locus.

Second, it is more challenging to build predictive models of resistance if each population has its own resistance markers. Metabolic resistance is often driven by increased expression of certain genes, and part of the reason for its multi-allelic nature is that there are many ways by which expression can be increased (copy number variants, mutation of *cis*-regulatory region, modified transposable element activity). To capture all of these in a single assay, it may be necessary to consider the possibility of transcriptomic or even proteomic surveillance[50]. This brings its own challenges. Unlike DNA, gene expression and protein synthesis are variable across tissues, life stages and environments, and also degrade quickly after death. Collection methodologies therefore need to be refined in order to achieve reliable and comparable results across studies. It is however becoming increasingly apparent that these challenges should be addressed and these avenues explored.

## Methods

### Sample collection and phenotyping
Mosquitoes were collected in 2017 and 2018 as larvae from six locations in West Africa (Benin: Avrankou [6.550, 2.667], Côte d'Ivoire: Aboisso [5.467, −3.200], Ghana: Madina [5.683, −0.166], Korle-Bu [5.537, −0.240] and Obuasi [6.200, −1.683]; Togo: Baguida [6.161, 1.314], Fig. 1). Larvae were collected by dipping from multiple habitats, pooled, and raised to adulthood in the laboratory and females were phenotyped for either deltamethrin or PM using a custom dose-response assay with WHO standard tubes, which was designed to identify the most resistant and susceptible individuals, while removing those of intermediate resistance. Non-blood-fed adult females were separated and at 3-5 days old bioassayed using WHO tubes in replicates of ~25 with either deltamethrin or pirimiphos methyl (PM) papers. Initially, to determine appropriate doses to produce well-separated phenotypic groups, which eliminated those of intermediate phenotype (Fig. S9), 1–2 replicate tubes were run (for 60 min, with mortality assessed 24 h later) at a range of concentrations reflecting X-fold the standard WHO diagnostic dose of 0.05% for deltamethrin (0.5X, 1X, 2X, 5X and 10X) and of 0.25% for PM (0.5X, 1X, 2X). From these preliminary data appropriate "lower" and "higher" doses (Fig. S9) were determined to achieve the desired phenotypic separation. Our initial plan to use different concentrations with a fixed (60 min) time was followed in all cases with the exception of Obuasi and Baguida phenotyped for PM, for which survivorship with 1X papers at 60 min was very low (≤2%, indicating susceptibility according to the WHO criterion). In these cases we obtained separated phenotypic groups by using a single concentration (0.5X) but varying exposure times. Bioassay doses used are shown in Table S1, and results from bioassays in Fig. S10. Note that with the exception of PM phenotypes in the two sites above, all populations tested conform to WHO defined resistance (Fig. S10 and Table S1), and in most cases for deltamethrin resistance would be classified as substantial (<90% at ≥5X). Samples were imported into the UK under import license IMP/GEN/2014/06 issued by the Department for Environment, Food and Rural Affairs (24th March 2014).

DNA was extracted from individual mosquitoes using nexttec extraction kits (Biotechnologie GmbH). Species identity was determined using two molecular methods designed to discriminate between *An. gambiae*, *An. coluzzii* and *An. arabiensis*: a PCR of species-specific SINE insertion polymorphisms as described in[51], and a melt curve analysis[52]. Breakdown of bioassay results by molecular species is

shown in Table S1. In two sites (Madina and Aboisso) *An. gambiae* and *An. coluzzii* were both present in the collections in substantial proportions and in each case *An. gambiae* were over-represented relative to *An. coluzzii* in resistant compared to susceptible groups (Madina, deltamethrin odds ratio = 19.2; PM odds ratio = 2.1; Aboisso, PM odds ratio = 36.8). This indicates that bioassay mortalities will be overestimated for *An. gambiae* in each case, compared to the results for *An. gambiae* s.l., this will apply to both the lower and higher doses. Therefore, pronounced phenotypic segregation between susceptible and resistant groups should still remain, though potentially quantitatively different from the *An. gambiae* s.l. data shown in Fig. S10. The complete list of specimens, sampling times and locations, and species assignments are available in Supplementary Data 1.

Sample sets (mosquitoes of a given species, from a given location, exposed to a given insecticides) to send for sequencing were chosen on the basis of sample size and to obtain a balance of deltamethrin/PM data. Final sample sizes for each sample set, after QC filtering of sequencing data, are shown in Table 1.

### Whole genome sequencing and bioinformatic analysis
Overall, 1258 samples from this study were whole-genome sequenced as part of the *Anopheles gambiae* 1000 genomes project (Ag1000G) release v3.2. Full details of library preparation, sequencing, alignment, SNP calling, CNV calling and phasing are detailed on the Ag1000G website (https://malariagen.github.io/vector-data/ag3/methods.html).

Briefly, individuals were sequenced to a target coverage of 30x on an Illumina HiSeq X, generating 150 bp paired-end reads. Reads were aligned to the AgamP4 reference genome using BWA, and indel realignment was performed using GATK version 3.7-0. Genotypes were called for each sample independently using GATK version 3.7-0 UnifiedGenotyper in genotyping mode, given all possible alleles at all genomic sites where the reference base was not "N". Sample QC removed 62 samples for low coverage (<10×), 215 samples for cross-contamination (alpha > 4.5%[53]) and 8 samples as apparent technical replicates (genetic distance below 0.006). 973 samples passed QC filtering. Sex was called using the modal coverage ratio between chromosomes X and 3 R (ratio between 0.4 and 0.6 = male, ratio between 0.8–1.2 = female, other ratios would lead to sample exclusion).

Known CNVs in specific genes of interest (*Cyp6aa1–Cyp6p2*, *Gstu4–Gste3*, *Cyp6m2–Cyp6m4*, Cyp6z3–*Cyp6z1*, *Cyp9k1*, *Ace1*) were detected using discordant reads associated with CNV alleles previously identified in Ag1000G release 3.0. Agnostic CNV detection was performed using normalised sequencing coverage calculated in 300 bp windows and a hidden Markov model (HMM) to estimate the copy number state at each window. This allows the detection of CNVs genome-wide, and of novel CNV alleles in the regions of interest. Gene copy number was calculated as the modal value of the HMM along each gene. A novel CNV in the regions of interest was identified if there was increased copy number according to modal coverage, but no discordant reads supporting the presence of known CNV alleles.

### Kinship analysis
We calculated pairwise kinship between all samples using the KING statistic[54] implemented in NGSRelate[55] using SNP data across the whole genome. Whole-genome SNPs were used because the recombination rate on such a small genome can lead to large disparity in kinship values between chromosomes (Fig. S11). Results indicated a slight positive bias in kinship, with the mode of the distribution slightly above 0 (Fig. S12). Because of this positive bias in kinship values, we sought to empirically establish the most parsimonious threshold to identify full siblings in our data, instead of the threshold of 0.177 suggested in the manual (https://www.kingrelatedness.com/manual.shtml). For all possible threshold between 0.15 and 0.35, in increments of 0.005, we identified all full siblings and counted the proportion of full sib groups that contained inconsistencies (where

siblings of siblings were not themselves classed as siblings). We chose the threshold 0.195 as that which produced the smallest proportion of inconsistent sib groups. We identified full siblings as any pair of individuals with a kinship value greater than this threshold,) and obtained full sib groups by considering that any siblings of siblings were themselves siblings. No full siblings were found between populations.

### Candidate marker and CNV association analysis

We investigated the association between phenotype and a range of known SNPs in candidate resistance genes (*Ace1*–280S all *Vgsc* SNPs reported in[21], *Rdl*–296G, *Rdl*–296S, *Cyp4j5*–43F, *Gste2*–114T and *Gste2*–119V) using generalised linear models in R, with binomial error and a logit link function, with phenotype as the dependent variable and SNP genotypes as independent variables, coded numerically as the number of mutant alleles (possible values of 0, 1 and 2). Within each sample set, we included all SNPs with an allele frequency of at least 10% in the analysis. We used a forward step-wise procedure, calculating the significance of adding each marker to the current model using the *anova* function. Starting from the null model, we added the most significant marker to the model and then repeated the process until no remaining markers provided a significant improvement. We used the same procedure to investigate the phenotypic association of gene copy number.

### Windowed measures of differentiation ($F_{ST}$, PBS, $H_{12}$)

$F_{ST}$ in 1000 SNP windows was calculated between resistant and susceptible samples within each sample set using the *moving_patterson_fst* function in *scikit-allel*, after filtering SNPs for missing data and accessibility[56] and removing singletons. In order to take full advantage of the full sample set despite the non-independence of full siblings, we performed up to 100 permutations in which one randomly chosen individual per sib group was used in the calculation of $F_{ST}$, and took the mean of all permutations. The set of possible permutations was sampled without replacement, such that in some sample sets fewer than 100 permutations were possible. We identified provisional windows of interest ("peaks") that were outliers in the data. We reasoned that true $F_{ST}$ peaks in these data should only be positive, meaning that the left hand side of the $F_{ST}$ distribution (Fig. S13) should be largely unaffected by the number of peaks in the data. We therefore used the left hand part of the $F_{ST}$ distribution to determine what the limits of the right hand part would typically look like in the absence of peaks, and identified outliers as positive windows beyond these limits. To do this, we took the difference between the smallest $F_{ST}$ value and the mode of the distribution, and considered an outlier to be any value more than three times this distance away from the mode on the right hand side (Fig. S13).

The existence of extended haplotype homozygosity in a region (due to a selective sweep) could cause a peak in windowed $F_{ST}$ even if the swept haplotype is unrelated to the phenotype, because non-independence of SNPs in the window would lead to increased variance in $F_{ST}$ compared to other genomic regions. This led, for example, to spurious provisional peaks in the *Ace1* region in sample sets phenotyped against deltamethrin (Fig. S5). To filter out these peaks, we performed 200 simulations in which the phenotype labels were randomly permuted and $F_{ST}$ recalculated as above. Provisional windows of interest were retained if their observed $F_{ST}$ was higher than the 99th centile of the simulations.

$H_{12}$ was calculated using phased biallelic SNPs in 1000 SNP windows, using the *garuds_h* function in scikit-allel. 200 phenotype permutations were performed as above. PBS was calculated using segregating SNPs in 1000 SNP windows using the *pbs* function in scikit-allel, after filtering SNPs for accessibility using the Ag1000G phase 3 gamb_colu site mask. For the outgroup for the PBS calculation, we used conspecific samples from Mali collected in 2004, available as part of the Ag1000G phase3 data release. For both $H_{12}$ and PBS, phenotype

permutations were performed as for $F_{ST}$ to filter out false positives caused by the presence of extended swept haplotypes.

### Haplotype association

Within each window of interest identified through the $F_{ST}$ analysis, we explored the presence of swept haplotypes that could be associated with phenotype. Haplotype clusters were determined by hierarchical clustering on pairwise genetic distance ($D_{xy}$) between haplotypes, and cutting the tree at a height of 0.001. Clusters larger than 20 haplotypes were tested for association with phenotype using a generalised linear model with binomial error and logit link function, with phenotype as the response and sample genotype (number of copies of the haplotype) as a numerical independent variable.

### Genome-wide association analysis

In the GWAS, a single permutation of sibling removal was randomly chosen for the analysis, since averaging over permutations would not produce interpretable $P$ values. In each sample set, all SNPs passing accessibility filters[56] with no missing data and a minor allele count of at least five were included in the analysis, including sites that failed accessibility filters.

In an original pass of the GWAS analysis, we found SNPs significantly correlated with phenotype, but which showed strong allelic imbalance and, in some cases, heterozygote excess. We found that they were the result of mis-alignment of *Anopheles* reads. Assembling and BLASTing the reads in question indicated that they likely originated from *Asaia* bacteria. There are two possible explanations for this. The first is that of a metagenomic association with resistance. The second is of sample contamination differentially affecting the two phenotypes. We investigated this possibility by examining the distribution of the levels of *Asaia* reads with respect to the position of the sample on the plates in which DNA was stored. *Asaia* reads were non-randomly distributed across plates (Fig. S14), indicating that they are likely to be the result of contamination, but creating the appearance of correlation with phenotype. To account for this in the GWAS, we used Bracken v2.5[57] with the GTDB database release 89 (https://data.ace.uq.edu.au/public/gtdb/data/releases/release89/) to estimate the amount of *Asaia* contamination in each sample and excluded SNP loci where genotype was correlated with *Asaia* levels ($P < 0.05$).

$P$-values of association with phenotype for each SNP were obtained by generalised linear modelling with binomial error and logit link function, with phenotype as the response variable and genotype (number of non-reference alleles) as a numeric response variable. False discovery rate correction was applied using the *fdrtool* package in R[58]. In several sample sets, no significant $P$ values remained after false discovery rate correction. Since recent selection on a SNP should lead to a broad selection signal, we took the 1000 most significant SNPs in each sample set and looked for 100,000 bp windows that contained at least 10 SNPs among the top 1000. Effects of individual SNPs were determined using SNPeff[59].

### GWAS sample size analysis

Although we found significant associations of individual SNPs in our analysis of established resistance markers for deltamethrin resistance, our agnostic SNP-level GWAS returned no markers passing FDR control. We therefore asked what sample size would have been required in order for these significant established markers to have been detected in the agnostic genome-wide analysis. Taking the *Vgsc*–1527T and *Cyp4j5*–43F loci in Avrankou, we used the observed allele frequencies in each of the resistant and susceptible subsamples to randomly generate new sample sets of a given size, split into 50/50 resistant/susceptible samples. We thus maintained the observed effect sizes for these loci, projected onto new sample sizes. For each random sample set, we calculated $P$ values following the same procedure as for our GWAS. FDR correction depends on the number of truly associated and

unassociated genes in the analysis; we therefore simulated 10 such loci that were truly associated with resistance across the genome (each drawn independently) and 7 million non-associated SNPs (approximately 7 million SNPs were used in the Avrankou GWAS), whose $P$ values were drawn from a random uniform distribution. This simulation was performed 500 times for each sample size.

## Reporting summary

Further information on research design is available in the Nature Portfolio Reporting Summary linked to this article.

## Data availability

The sequencing data generated in this study have been deposited in the ENA short read archive database, with all accession codes provided in Supplementary Data 1. The accession number for the genome assembly to which the reads were aligned is GCA_000005575.1 (https://www.ebi.ac.uk/ena/browser/view/GCA_000005575.1). The processed SNP and CNV calling data were generated as part of the *Anopheles gambiae* 1000 genomes project v3.2 and are available at https://www.malariagen.net/data. The bioassay data generated in this study are provided in Supplementary Data 1.

## Code availability

Code used to analyse the data can be found in the github repository https://github.com/vigg-lstm/GAARD_work[17]. All sequencing, alignment, SNP and CNV calling was carried out as part of the *Anopheles gambiae* 1000 genomes project v3.2 (https://www.malariagen.net/data).

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

## Acknowledgements

This work was supported by the National Institute of Allergy and Infectious Diseases (NIAID R01-AI116811 to M.J.D. and D.W.) and the Medical Research Council (MR/T001070/1 to M.J.D., D.W. and E.R.L., MR/P02520X/1 to M.J.D. and D.W.). The latter grant is a UK-funded award and is part of the EDCTP2 programme supported by the European Union. M.J.D. is supported by a Royal Society Wolfson Fellowship (RSWF\FT\180003). We thank the *Anopheles gambiae* 1000 genomes project for carrying out the sequencing, quality control, SNP calling and for haplotype phasing the sequencing data and Luciene Salas Jennings and Andrew Carey for providing administrative support to the project.

## Author contributions

J.E., A.E.-Y., S.D., J.C., L.S.D., A.A.M., C.V.E., G.K.K. and B.G.K. collected the samples. E.J.R., D.P. and A.E.V.H. performed the lab work. E.R.L., S.C.N., A.M., N.J.H., C.S.C. and L.T.C. analysed the data. A.M. and D.P.K. oversaw the sample sequencing. D.W., M.J.D., A.E.-Y., L.S.D. and C.V.E. designed the study. E.R.L., S.C.N., N.A.D., M.J.D. and D.W. wrote the manuscript.

## Competing interests

The authors declare no competing interests.

## Additional information

[1]Department of Vector Biology, Liverpool School of Tropical Medicine, Pembroke Place, Liverpool L3 5QA, UK. [2]Department of Biomedical Sciences, University of Cape Coast, Cape Coast, Ghana. [3]Department of Parasitology, Noguchi Memorial Institute for Medical Research, University of Ghana, Accra, Ghana. [4]Tropical Infectious Diseases Research Centre (TIDRC), Université d'Abomey-Calavi (UAC), 01 B.P. 526 Cotonou, Benin. [5]Centre Suisse de Recherches Scientifiques en Côte d'Ivoire, 01 BP 1303 Abidjan, Côte d'Ivoire. [6]Laboratory of Ecology and Ecotoxicology, Department of Zoology, Faculty of Sciences, Université de Lomé, 01 B.P. 1515 Lomé, Togo. [7]Biology Centre of the Czech Academy of Sciences, Institute of Entomology, Branišovská 31, 370 05 České Budějovice, Czech Republic. [8]Big Data Institute, Li Ka Shing Centre for Health Information and Discovery, University of Oxford, Oxford, UK. [9]Wellcome Sanger Institute, Hinxton, Cambridge CB10 1SA, UK. [10]Deceased: Dominic P. Kwiatkowski ✉e-mail: eric.lucas@lstmed.ac.uk; martin.donnelly@lstmed.ac.uk; david.weetman@lstmed.ac.uk

