## [Peer Review File · Nature Communications]

Genome-wide association studies reveal novel loci associated with pyrethroid and organophosphate resistance in *Anopheles gambiae* and *Anopheles coluzzii*REVIEWERS' COMMENTS

Reviewer #1 (Remarks to the Author):

The title of the manuscript should be changed to reflect that the analysis is done in both *Anopheles gambiae* and *Anopheles coluzzii*. Both are referred to in the text.

Insecticide resistance is a major challenge to controlling mosquitoes in Africa. The use of insecticides is expected to continue for the foreseeable future so understanding their effect on populations of anophelines is important. This paper reports the molecular basis of resistance to two common insecticides deployed in Africa with 10 sites sampled from four countries in western Africa. The authors were interested in establishing the allelic basis of resistance across these populations (both at target and non-target genomic sites looking at single nucleotide polymorphisms and target gene copy numbers). The bioinformatic analyses, including GWAS analyses are appropriate and thorough and I believe the sample sizes warrant the conclusions made. The most surprising of these is that the level of allelic variation contributing to resistance separate from the known genetic targets (i.e. known resistance alleles that are the major contributors to resistance) varies considerably across geographical locations. This finding will impact resistance management programs and, combined with the quality of the experimental design and analysis, justifies publication.

Reviewer #2 (Remarks to the Author):

While a huge number of studies have now investigated the molecular basis of insecticide resistance – the number of studies employing a genome-wide approach remain surprisingly small, particularly in the case of non-model organisms. This study goes some way to addressing this deficit by conducting a multi-country genome-wide association study of resistance to two insecticides in *Anopheles gambiae*. The authors analyses reveal a complex pattern of resistance implicating multiple genes, both previously described and novel. The data presented is novel and interesting as it moves beyond the current well-characterised resistance genes of large effect by providing new insights into the genetic factors that explain residual resistance. I found the manuscript to be very well written, with the results intelligently and thoughtfully discussed. I am enthusiastic about the study and have only two minor suggestions for the authors to consider.

P12 – In the discussion of variation in the resistance phenotype conferred by target-site resistance mutations when in different genetic backgrounds it may be useful to introduce the concept of epistasis.

Discussion – one aspect of this study that is very exciting is the opportunity it offers to revisit the longstanding question of the frequency with which resistance is monogenic vs. polygenic based. Theory suggests that selection within a continuous phenotypic distribution, such as a small laboratory population, typically selects for polygenic resistance, in contrast to selection for phenotypes outside the normal phenotypic distribution in the field which favours a monogenic response (DOI: 10.1534/genetics.112.141895). The current work clearly offers an opportunity to inform this debate and I ask the authors to consider adding some concise discussion on this point in the light of their results.

RESPONSE TO REVIEWERS' COMMENTS

>>> We thank the reviewers for their positive and helpful comments. Please find out point-by-point answers below.

Reviewer #1 (Remarks to the Author):

The title of the manuscript should be changed to reflect that the analysis is done in both *Anopheles gambiae* and *Anopheles coluzzii*. Both are referred to in the text.

>>> We have changed this.

Insecticide resistance is a major challenge to controlling mosquitoes in Africa. The use of insecticides is expected to continue for the foreseeable future so understanding their effect on populations of anophelines is important. This paper reports the molecular basis of resistance to two common insecticides deployed in Africa with 10 sites sampled from four countries in western Africa. The authors were interested in establishing the allelic basis of resistance across these populations (both at target and non-target genomic sites looking at single nucleotide polymorphisms and target gene copy numbers). The bioinformatic analyses, including GWAS analyses are appropriate and thorough and I believe the sample sizes warrant the conclusions made. The most surprising of these is that the level of allelic variation contributing to resistance separate from the known genetic targets (i.e. known resistance alleles that are the major contributors to resistance) varies considerably across geographical locations. This finding will impact resistance management programs and, combined with the quality of the experimental design and analysis, justifies publication.

>>> We are grateful to the reviewer for this positive assessment.

Reviewer #2 (Remarks to the Author):

While a huge number of studies have now investigated the molecular basis of insecticide resistance – the number of studies employing a genome-wide approach remain surprisingly small, particularly in the case of non-model organisms. This study goes some way to addressing this deficit by conducting a multi-country genome-wide association study of resistance to two insecticides in *Anopheles gambiae*. The authors analyses reveal a complex pattern of resistance implicating multiple genes, both previously described and novel. The data presented is novel and interesting as it moves beyond the current well-characterised resistance genes of large effect by providing new insights into the genetic factors that explain residual resistance. I found the manuscript to be very well written, with the results intelligently and thoughtfully discussed. I am enthusiastic about the study and have only two minor suggestions for the authors to consider.

>>> Thank you for this positive review.

P12 – In the discussion of variation in the resistance phenotype conferred by target-site resistance mutations when in different genetic backgrounds it may be useful to introduce the concept of epistasis.

>>> Thank you for this suggestion. We have added this on lines 287-289.

Discussion – one aspect of this study that is very exciting is the opportunity it offers to revisit the longstanding question of the frequency with which resistance is monogenic vs. polygenic based. Theory suggests that selection within a continuous phenotypic distribution, such as a small laboratory population, typically selects for polygenic resistance, in contrast to selection for

phenotypes outside the normal phenotypic distribution in the field which favours a monogenic response (DOI: 10.1534/genetics.112.141895). The current work clearly offers an opportunity to inform this debate and I ask the authors to consider adding some concise discussion on this point in the light of their results.

>>> Thank you for this interesting addition to the discussion, which we have incorporated on lines 404-421.